# First Characterization of a Cyanobacterial Xi-Class Glutathione S-Transferase in *Synechocystis* PCC 6803

**DOI:** 10.3390/antiox13121577

**Published:** 2024-12-20

**Authors:** Fanny Marceau, Marlène Lamothe-Sibold, Sandrine Farci, Soufian Ouchane, Corinne Cassier-Chauvat, Franck Chauvat

**Affiliations:** Université Paris-Saclay, CEA, CNRS, Institute for Integrative Biology of the Cell (I2BC), 91198 Gif-sur-Yvette, France; fanny.marceau@saint-gobain.com (F.M.); marlene.lamothe-sibold@i2bc.paris-saclay.fr (M.L.-S.); sandrine.farci@cea.fr (S.F.); soufian.ouchane@i2bc.paris-saclay.fr (S.O.); corinne.cassier-chauvat@cea.fr (C.C.-C.)

**Keywords:** cyanobacteria, catalase, benzoquinone, cobalt, detoxification, glutathione S-transferase, glutathionyl–hydroquinone reductase, iron homeostasis, iron–sulfur cluster, oxidative stress, ROS

## Abstract

Glutathione S-transferases (GSTs) are evolutionarily conserved enzymes crucial for cell detoxication. They are viewed as having evolved in cyanobacteria, the ancient photosynthetic prokaryotes that colonize our planet and play a crucial role for its biosphere. Xi-class GSTs, characterized by their specific glutathionyl–hydroquinone reductase activity, have been observed in prokaryotes, fungi and plants, but have not yet been studied in cyanobacteria. In this study, we have analyzed the presumptive Xi-class GST, designated as Slr0605, of the unicellular model cyanobacterium *Synechocystis* PCC 6803. We report that Slr0605 is a homodimeric protein that has genuine glutathionyl–hydroquinone reductase activity. Though Slr0605 is not essential for cell growth under standard photoautotrophic conditions, it plays a prominent role in the protection against not only benzoquinone, but also cobalt-excess stress. Indeed, Slr0605 acts in defense against the cobalt-elicited disturbances of iron homeostasis, iron–sulfur cluster repair, catalase activity and the level of reactive oxygen species, which are all crucial for cell life.

## 1. Introduction

Glutathione S-transferases (GSTs, EC 2.5.1.18) constitute a superfamily of evolutionarily conserved enzymes acting in cell detoxication [1], which have great importance for human health [2,3] and agriculture [4,5]. They conjugate the crucial antioxidant metabolite glutathione (γ-glutamyl-cysteinyl-glycine, GSH) on many endogenous or exogenous molecules (metabolite by-products, chemicals, metals, oxidants), which are then detoxified and/or eliminated [6,7,8,9]. GSTs can also modulate protein activity by glutathionylation/deglutathionylation (the formation/reduction of a disulfide bridge between the cysteinyl residue of GSH and the cysteinyl residue of a target protein) [10,11].

Glutathione S-transferases are commonly divided into various families named with Greek letters: alpha, beta, chi, …, xi and zeta. Xi-class GSTs have been recently identified as being specifically endowed with glutathionyl–hydroquinone reductase activity. They are widely distributed in prokaryotes, fungi and plants [12,13,14], but their role in stress tolerance has been overlooked. Furthermore, Xi-class GSTs have not been studied in cyanobacteria, although these photosynthetic prokaryotes are regarded as the originators of GSH-dependent enzymes that protect themselves from the metabolic and environmental stresses they frequently experience [1,15,16]. Furthermore, in colonizing our planet, cyanobacteria are very important organisms that produce a large amount of biomass and oxygen for our food chain [17]. Hence, they constitute the first biological barrier against the entry of pollutants, like heavy metals, into our food web, which are increasingly being released into the environment by natural sources (volcanoes or forest fires) and anthropogenic activities (mining, fossil-fuels burning, metallurgy, etc.) [18]. Moreover, cyanobacteria have great biotechnological potential for bioremediation and the bioproduction of chemicals from solar energy, water (fresh and/or marine) and CO_2_ [19,20].

The model unicellular cyanobacterium *Synechocystis* PCC 6803 (hereafter *Synechocystis*) is suitable for in vivo analysis of GSTs, with its small (4.0 Mb) manipulable genome encoding six evolutionarily conserved GSTs, designated as Sll0067, Sll1147, Sll1545, Sll1902, Slr0236 and Slr0605 [1,21]. Both Sll1545 and Slr0236 were found to contribute to protection against photo-oxidative stress [21,22,23], while Sll1147 was shown to contribute to resistance to heat and lipid peroxidation [24]. Sll0067 appeared to act in the detoxication of isothiocyanates [25] and methylglyoxal [6,26], the latter of which is a metabolic by-product causing diabetes in humans [1] and age-related disorders [27].

In this study, we have analyzed the role of the presumptive Xi-class GST, Slr0605, using deletion/complementation, purification from *Synechocystis* and enzyme assays. We report that Slr0605 plays a role in defending against excess cobalt (Co), a metal required in trace amounts for the synthesis of vitamin B12 and other cobalamins [28,29,30], which can become toxic at high doses [31]. We also show that Slr0605 is a true Xi-class GST, endowed with a genuine dimeric form and glutathionyl–hydroquinone reductase activity. Furthermore, Slr0605 plays an important role in protecting against Co-elicited disturbances of iron (Fe) homeostasis and the biogenesis/repair of enzymatic iron–sulfur [Fe-S] clusters, which trigger oxidative stress.

## 2. Materials and Methods

### 2.1. Bacterial Strains, Growth Conditions and Gene-Transfer Procedures

*Escherichia coli* (*E. coli*) strains (Appendix A) used for gene manipulations (TOP10 and NEB10 beta) or the conjugative transfer (CM404) of the RSF1010-derived replicative plasmids to *Synechocystis* PCC 6803 (*Synechocystis*) were grown at 37 °C (TOP10 and NEB10 beta) or at 30 °C (CM404) on LB medium containing appropriate antibiotics: ampicillin (Ap) 100 μg·mL^−1^, kanamycin (Km) 50 μg·mL^−1^, streptomycin (Sm) 25 μg·mL^−1^ or spectinomycin (Sp) 75 μg·mL^−1^. *Synechocystis* was grown at 30 °C, under continuous white light (cool white Sylvania Luxline Plus; 1100 lux; 14 μE m^−2^ s^−1^ or 7500 lux; 94 μE m^−2^ s^−1^) in liquid mineral medium (MM), i.e., BG11 [32] enriched with 3.78 mM Na_2_CO_3_, under continuous agitation (140 rpm, Infors rotary shaker, Basel, Switzerland). For some experiments, Co or p-benzoquinone (Sigma-Aldrich, Saint Quentin-Falavier, France) were added to MM at the indicated concentrations. Growth was monitored by regular measurements of optical density at 750 nm (OD_750_; 1 OD_750_ = 5 × 10^7^ cell.mL^−1^). In some cases, exponentially growing cells were serially diluted (5-fold) in MM, spread on MM (with or without the indicated metals) solidified with 10 g·L^−1^ Bacto Agar (Difco) and incubated for several days under photoautotrophic conditions until photography and/or analysis. The DNA cassettes for the targeted deletion of sufR and slr0605 were introduced in *Synechocystis* by transformation [33]. The pCK-derived plasmids for the high-level expression of slr0605 (pCKslr0605) or its derivative (pCKslr0605strep), translationally fused to the Strep-tagII sequence for easy protein purification (Appendix A), were introduced in *Synechocystis* by conjugation [34] using a 24 h co-incubation of *E. coli* and *Synechocystis* cells. Transformants and conjugants were plated on solid MM. The antibiotics used for selection were kanamycin (Km) 50–300 μg·mL^−1^, spectinomycin (Sp) 5 μg·mL^−1^ and streptomycin (Sm) 5 μg·mL^−1^. The integration of the slr0605 deletion cassette into the *Synechocystis* chromosome and the presence of the slr0605 expression plasmids were verified by PCR and DNA sequencing (Mix2Seq Kit, Eurofins Genomics, Nantes, France) using appropriate oligonucleotide primers (Appendix A).

### 2.2. Construction of ΔsufR::Km^R^ Deletion Cassette for Targeted Deletion of sufR Gene

The ΔsufR::Km^R^ deletion cassette was constructed by replacing the sll0088 coding sequence (CS, from codon 8 to codon 192) with a transcription-terminator-less kanamycin resistance gene (Km^R^) for selection, while preserving the sll0088 flanking regions for homologous recombination mediating targeted gene replacement in *Synechocystis* [33]. These DNA regions (about 300 bp) were PCR amplified from the *Synechocystis* chromosome with specific primers (Appendix A) and joined by PCR-driven overlap extension in a single DNA segment harboring a SmaI restriction site in place of the sufR CS (Appendix A). After cloning in pGEM-T, the resulting plasmid was opened at its unique SmaI site, where we cloned the transcription-terminator-less Km^R^ marker (a HincII segment from pUC4K plasmid) in the same orientation as the sufR CS that it replaced (Appendix A). The ΔsufR::Km^R^ resulting plasmid was verified by PCR (Appendix A) and DNA sequencing (Mix2Seq Kit, Eurofins Genomics). It was then transformed to *Synechocystis*, where a double crossing-over occurring in the sufR flanking regions integrated the Km^R^ marker in place of the sufR CS in all copies of the polyploid [33] *Synechocystis* chromosome. The absence of WT chromosome copies in the ΔsufR::Km^R^ mutant was confirmed by analyzing cells grown for multiple generations in the absence of Km to allow the propagation of WT (sufR^+^, Km^S^) chromosome copies which could have escaped PCR detection. As expected, the ΔsufR::Km^R^ mutant possessed only sufR::Km^R^ chromosomes (Appendix A, see the 580 bp and 720 bp products from PCR1 and PCR2, respectively), and WT chromosomes (with the absence of 538 bp and 783 bp bands from PCR3 and PCR4, respectively).

### 2.3. Construction of Δslr0605::Km^R^ Deletion Cassette for Targeted Deletion of slr0605 Gene

The two *Synechocystis* chromosomal DNA regions flanking the slr0605 coding sequence (CS) were amplified by PCR (Phusion, Invitrogen, ThermoFisher, Ottawa, ON, Canada) from the *Synechocystis* chromosome, using specific oligonucleotide primers (Appendix A). In parallel, the transcription-terminator-less Sm^R^/Sp^R^ marker gene was PCR amplified from the pFC1 plasmid [34] using specific primers (Appendix A). These three PCR products were assembled (NEBuilder^®^ HiFi DNA Assembly) with the pGEM-T plasmid (Promega, Charbonnière-les-bains, France). The Δslr0605::Sm^R^/Sp^R^ resulting plasmid was verified by PCR (Appendix A) and nucleotide sequencing (Mix2Seq Kit, Eurofins Genomics). It was then transformed to *Synechocystis*, where homologous recombinations occurring in the slr0605 flanking regions replaced the slr0605 CS by the Sm^R^/Sp^R^ marker in all copies of the *Synechocystis* chromosome (Appendix A). We verified that the Δslr0605:Sm^R^/Sp^R^ mutant possesses only Δslr0605::Sm^R^/Sp^R^ mutant chromosomes, even when growing in the absence of Sm and Sp, to stop the counter-selection of possible WT chromosome copies that could have escaped PCR detection.

### 2.4. Construction of pCKslr0605 and pCKslr0605 Plasmids Expressing slr0605 Gene, or Its Strep-Tagged Derivative, Respectively

The slr0605 coding sequence was PCR amplified from *Synechocystis* DNA with specific oligonucleotide primers (Appendix A) that introduced an NdeI restriction site upstream of its start codon and an EcoRI site downstream of its stop codon. This PCR product was cut with both NdeI and EcoRI and cloned in the pCK plasmid opened with the same enzymes, yielding the pCKslr0605 plasmid (Appendix A). Similarly, the slr0605 gene, translationally fused at its 3′OH side to the Strep-tagII for the facile purification of the Slr0605 protein, was also cloned in pCK, yielding the pCKslr0605 plasmid.

### 2.5. Pigment Extraction and Quantification

*Synechocystis* cells incubated on plates were re-suspended in 1 mL of ultra-pure water (A_750_ nm = 0.2), centrifuged (10 min, 14,000 rpm at 4 °C) and the pellet was carefully resuspended in 1 mL of cooled methanol (4 °C). The absorbance of the solution was then measured at 470, 665 and 720 nm, and pigment concentrations were calculated using the following equations: chlorophyll a concentration (µg/mL): [Chl a] = 12.9447 (A_665nm_ − A_720nm_) [35]; total carotenoids in µg/mL = 1000 (A_470nm_ − A_720nm_) − 2.86 ([Chl *a*]/221) [36]. To measure the phycocyanin content, the 1 mL cell suspension in ultra-pure water was heated for 8 min at 75 °C, and the phycocyanin concentration was estimated as follows: [PC] = (A_620nm_ − A_750nm_) − (A_620nm_ heated − A_750nm_ heated) [37].

### 2.6. Reactive Oxygen Species (ROS) Assay

The content of ROS (H_2_O_2_, HO• and ROO•) was measured as previously described [21], using the standard probe 2′,7′-dichlorodihydrofluorescein diacetate (DCFH-DA; Sigma-Aldrich) that is oxidized into the fluorescent derivative dichlorofluorescein (DCF) by the sequential action of cellular esterases and ROS [38]. Briefly, WT and mutant cells incubated under standard or stress conditions (OD_750_ = 0.5) were collected by a 10 min centrifugation at 5000 rpm and resuspended in fresh MM (2.5 × 10^7^ cells.mL^−1^). They were incubated in the dark for 1 h at 30 °C in the presence of 5 μM DCFH-DA, solubilized in ethanol. Then, the fluorescence excitation/emission, normalized on the basis of the OD_750_, was read at λ_exc_/λ_em_ = 485/520 nm at 30 °C using a microplate reader (CLARIOstar; BMG LABTECH, Champigny sur Marne, France).

### 2.7. Protein Extraction and Purification

Multiple 50 mL cultures of the *Synechocystis* strain harboring the pCKslr0605strep plasmid (OD_750_ = 0.5; 1.25 × 10^9^ cells) were harvested by centrifugation (7000× *g* for 5 min at 4 °C) and resuspended in 1.5 mL of 50 mM Tris-HCl buffer at pH 7.6. Cells were immediately frozen in an Eaton press chamber pre-cooled in a dry-ice ethanol bath and disrupted (250 MPa). Cell debris were eliminated by centrifugation (10 min, 14,000 rpm at 4 °C). The supernatant was ultracentrifuged (10 min, 60,000 rpm at 4 °C in TLA-100 rotor) to separate soluble and insoluble fractions. Protein concentration was measured by the Bradford assay (Biorad, Marnes-la-Coquette, France), using BSA (bovine serum albumin) as the standard. Proteins of 5 to 10 μg were separated onto a 12% (*w*/*v*) SDS-PAGE (stacking gel: 4% acrylamide-bis-acrylamide, 125 mM Tris/HCl pH 6.8, 8.33% glycerol, 0.21% SDS; running gel: 12% acrylamide-bis-acrylamide, 375 mM Tris/HCl pH 8.8, 10% glycerol, 0.02% SDS), run for 2 h at 100 V under denaturing conditions (40 mM Tris, 300 mM glycine, 0.1% SDS). The gels were either stained with 0.25% Coomassie blue, 45% ethanol, or 9% acetic acid and then discolored (25% ethanol, 10% acetic acid), or analyzed by Western blotting. The blue prestained protein ladder (Sigma Aldrich) was used as the molecular weight marker.

The Slr0605–Strep-tagged protein was purified from 100 mL of liquid culture at 2.0 OD_750_ = 2; 5 × 10^9^ cells. After ultracentrifugation, the soluble fraction was diluted in 4 mL of binding buffer (100 mM Tris-HCl, 150 mM NaCl, 1 mM EDTA, pH 8). The Slr0605–Strep-tagged protein was purified using 1 mL of Strep-Tactin XT Sepharose (Cytiva, Velizy Villacoublay, France) and eluted, according to the supplier’s recommendation, in the presence of 50 mM biotin. Then, Slr0605-strep was concentrated 80-fold using the Amicon^®^ Ultra-15 10K kit (Merck, Guyancourt, France) to eliminate of molecules with a mass <10 kDa. The absorbance at 280 nm was measured using Nanodrop^®^ (ThermoFisher, Ottawa, ON, Canada). The concentration of the purified recombinant proteins was calculated using the molecular mass (37,665.53 g.mol^−1^) and molar absorption coefficients (88,935 M^−1^.cm^−1^) calculated by Expasy ProtParam [39].

### 2.8. Western Blot Analysis

After SDS-PAGE, proteins were transferred onto a Polyvinylidene Difluoride membrane (PVDF, GE Healthcare, Amersham^TM^, Les Ulis, France) 1 h at 100 V (transfer buffer: 25 mM Tris, 192 mM glycine, 0.1% (*w*/*v*) SDS, 20% (*v*/*v*) ethanol) at room temperature (T°). The membrane was saturated for 2 h at room T° in the TBST buffer (50 mM Tris, 150 mM NaCl, 0.1% (*v*/*v*) Tween 20, 5% (*w*/*v*) BSA, pH 7.4). The membrane was probed for 2 h at room T° with primary mouse antibodies raised against the PsbA (Agrisera, 1:10,000 dilution), PsaA (Agrisera, 1:10,000 dilution) proteins or Strep-tagII (Sigma; 1:5000 dilution in TBST with BSA (*w*/*v*) 1%). The membrane was washed 5 times for 5 min in TBST and incubated with secondary horseradish peroxidase (HRP)-conjugated rabbit anti-mouse antibodies (Sigma; 1:5000 dilution). The membrane was washed 5 times for 5 min in TBST. Immune complexes were detected using a chemiluminescent HRP substrate, as described [40]. For accurate quantification, various protein amounts were migrated onto the gel, and immune complexes band intensity was measured using ImageJ 1.x software (densitometry measurement). The relative band intensity was calculated using the value obtained for the wild-type (WT) strain incubated under standard conditions as a reference.

### 2.9. Catalase Assay

It was performed as described [41]. Briefly, cell-free lysates containing 10 μg of total proteins from WT and Δslr0605 mutant strains were separated on a non-denaturating 12% (*w*/*v*) polyacrylamide gel. Then, the gel was washed 3 times for 10 min at room T° in ultra-pure water, with gentle agitation. It was then incubated for 10 min in a 0.01% (*v*/*v*) H_2_O_2_ solution, rinsed with ultra-pure water and soaked in a freshly-prepared solution containing 1% (*w*/*v*) FeCl_3_ and 1% (*w*/*v*) K_3_Fe(CN)_6_. The appearance of a high molecular weight achromic band resulting from catalase activity (decomposition of H_2_O_2_) was observed on a blue-green background (Prussian Blue). The reaction was stopped by washing in ultra-pure water to remove the ferric solution. Subsequently, the gel was scanned, and the intensity of the catalase activity band was measured using ImageJ 1.x software.

### 2.10. Superoxide Dismutase Assay

Total protein extracts (10 μg) were immediately mixed with a non-SDS/DTT protein dye and migrated on a non-denaturing 12% (*w*/*v*) acrylamide gel at 100 V until the protein dye reached the end of the gel. Then, the gel was incubated for 15 min at room T° in a Nitro Blue Tetrazolium chloride (NBT, ThermoFisher Scientific, Ottawa, ON, Canada) solution (50 mM Tris/HCl pH 7.6, 2.43 mM NBT in darkness). After the addition of 0.028 mM Riboflavin (Sigma Aldrich) and 28.48 μM TEMED (MP medicals, Illkirch France), the gel was incubated for 15 min under mild agitation. The Superoxide dismutase (SOD) reaction was then performed under white light (7000 lux) for 30 min at 30 °C. Riboflavin and light generate superoxide ions that are reduced by NBT, thereby coloring the gel purple (formazan production), with the exception of the band corresponding to the SOD protein, which remains colorless. Subsequently, the gel was scanned, and the intensity of the SOD activity bands was measured using ImageJ software.

### 2.11. Measurement of Glutathionyl–Hydroquinone Reductase Activity

The GHR activity was evaluated as described [42]. Briefly, an absorption spectrum (230 to 400 nm) of 10 µM p-benzoquinone (Sigma-Aldrich) in 50 mM Tris-HCl buffer pH 7.5, was first recorded as the baseline of the assay. After 2 min, 50 µM reduced glutathione (Sigma-Aldrich) was added to the reaction mixture, and a new absorption spectrum was recorded. Reduced glutathione reacts rapidly with p-benzoquinone, inducing an absorption peak shift from 247 nm to 300 nm (with the formation of S-Glutathionyl–p-hydrobenzoquinone). Finally, 1.7 µg of Slr0605-Strep-tagged purified protein was added to the reaction mixture, and an absorption spectrum was recorded every min for 8 min. A peak shift from 300 nm to 285 nm was observed over time, testifying the synthesis of the p-benzohydroquinone product by the enzyme.

### 2.12. Statistical Analyses

All experiments were carried out in at least triplicate (n = 3), and the results are expressed as the mean ± standard deviation (SD). Statistical analysis was conducted using one-way analysis of variance (ANOVA) or the Welch’s *t*-test. Statistical significance was determined at a level of *p* < 0.05 (symbolized by *), *p* < 0.005 (**), *p* < 0.0005 (***) or *p* < 0.00005 (****).

All statistical tests were performed using GraphPad Prism 10 software. The Welch’s *t*-test [43] was used to analyze the growth and pigment assay data from independent experiments and the significance of ROS assay results (comparison of only two independent groups). One-factor ANOVA and the Tukey test were used to analyze the significance of (i) the comparison of catalase and superoxide dismutase activities in two strains incubated under various conditions, and (ii) the effect of iron supplementation observed for the same experiment (for the same batch of culture medium, same precultures, etc.). The Brown–Forsythe and Dunnett tests were used to analyze the results of western blots from different dilutions of protein extracts [44].

## 3. Results and Discussion

### 3.1. Synechocystis PCC 6803 Is Particularly Sensitive to Cobalt That Reduces Chlorophyll Content

In the frame of our interest in the resistance of cyanobacteria to environmental stresses [18], we tested the influence of various metals on the growth of the model strain *Synechocystis* PCC 6803 (hereafter *Synechocystis*). The data showed that both cell growth and their content in photosynthetic pigments were altered in response to metal excess, especially Co (Figure 1). This finding confirms and extends previous reports on cobalt toxicity in *Synechocystis* [18,31,45] in showing that it is more affected by Co than by other metals. The most significant Co effects on the content of photosynthetic pigments, evaluated using the absorption spectra of extracted pigments, was the decrease in both carotenoids (decrease in absorbance in the range 400–490 nm) and chlorophyll a (decrease in absorption peaks at 435 nm and 665 nm) (Figure 1). The latter finding is consistent with previous reports that Co reduces the chlorophyll content of the filamentous cyanobacterium *Spirulina platensis* [46], and the unicellular species *Anacystis nidulans* UTEX 625 [47] and *Synechococcus* PCC 7942 [48], which are closely related [49].

### 3.2. Iron Supplementation Protects Synechocystis Against Cobalt-Elicited Declines of Growth and Pigments

Cobalt toxicity has been well studied in *E. coli*, where it was found to impair iron (Fe) homeostasis and the iron–sulfur [Fe-S] cluster of numerous enzymes, thereby triggering oxidative stress [30,50]. In cyanobacteria, iron is especially important [51]. Indeed, the iron quota (atoms per cell) in *Synechocystis* is one order of magnitude higher than in *E. coli* [52]. For example, the photosynthetic apparatus employs twelve Fe atoms for photosystem I (PSI), and ten Fe atoms for PSII, cytochrome b6f and cytochrome c-553 [53]. Furthermore, PSI harbors three [4Fe-4S] clusters: the F_A_ and F_B_ clusters in the PsaC subunit and the F_X_ cluster ligated to the PsaA and PsaB subunits [53]. Consequently, we have tested the influence of Fe supplementation on the growth and pigment content of *Synechocystis* exposed to Co. For an unclear reason, somehow related to Co, the pigment extraction data showing the Co-induced net decrease in chlorophyll a (absorption peak at 680–700 nm), a low decrease in carotenoids, and a small increase in phycocyanin (peak at 620 nm) (Figure 2) fit better with the whole-cell absorption spectra normalized at 569 nm (a valley of absorbance) than at 800 nm (a commonly-used value) (Figure 2). The discrepancy between the whole-cell spectra normalized at 569 nm or 800 nm was mainly observed for cells exposed to Co, not Fe. Collectively, the results showed that the Co-induced decline of cell growth and chlorophyll content (Figure 1) was attenuated by the presence of extra Fe (Figure 2).

### 3.3. Activation of Iron–Sulfur Cluster Repair Genes Alleviates Cobalt Toxicity

As it has been shown in *E. coli* that mutants lacking the Fe-S cluster assembling Suf machinery are hypersensitive to Co [54], we have tested the influence of the *suf* genes of *Synechocystis* on its tolerance to Co. In *Synechocystis*, the *suf* genes are grouped in the *sufBCDS* operon that is negatively regulated by the SufR repressor encoded by the *sll0088* gene [55]. We reasoned that a *sll0088* deletion mutant overexpressing the *sufBCDS* genes should be resistant to Co. Therefore, a Δ*sufR*::Km^R^ deletion cassette was constructed (Materials and Methods) by replacing the *sll0088* coding sequence (CS, from codon 8 to codon 192) by a transcription-terminator-less kanamycin resistance gene (Km^R^) for selection, while preserving the *sll0088* flanking regions for homologous recombination mediating targeted gene replacement in *Synechocystis* [33]. The Δ*sufR*::Km^R^ mutant, hereafter Δ*sufR*, grew as healthy as the wild-type (WT) strain under photoautotrophic conditions, showing that SufR is not essential to cell life in *Synechocystis*, as previously observed [53]. The Δ*sufR* mutant appeared to be more resistant to Co than the WT strain (Figure 3), unlike the previously described GST mutants that were used as controls: Δ*sll1545* [21], Δ*sll1147* [24], Δ*sll0067* [6], Δ*sll1902* (to be published elsewhere) and the presently constructed GST-less mutant Δ*slr0605* (see below).

### 3.4. Slr0605 GST Contributes to Cobalt Resistance in Synechocystis

Because GST can act in the protection against metal [7], and the role of Xi GST is poorly known, we tested the possible influence of the Slr0605 GST on resistance to cobalt. For this purpose, a Δ*slr0605*::Sm^R^/Sp^R^ deletion mutant was constructed (Section 2) by replacing the *slr0605* coding sequence with a transcription-terminator-less streptomycin and spectinomycin resistance gene (Sm^R^/Sp^R^) for selection. The Δ*slr0605*::Sm^R^/Sp^R^ mutant grew as fit as the wild-type (WT) strain under photoautotrophic conditions (Figure 4), demonstrating that Slr0605 is not essential to cell life, similar to other *Synechocystis* GSTs, such as Slr0236 [21], Sll1147 [24] and Sll0067 [6], unlike Sll1545, which is vital [21].

The possible influence of Slr0605 on tolerance to Co was tested by growing the WT and the Δ*slr0605*::Sm^R^/Sp^R^ mutant on liquid or solid medium with or without Co supplied as CoCl_2_ (Figure 4). The data showed that the Δ*slr0605*::Sm^R^/Sp^R^ mutant was more sensitive to Co than the WT strain. Pigment extraction analysis showed that the Co-elicited decrease in chlorophyll was exacerbated in the Δ*slr0605*::Sm^R^/Sp^R^ mutant compared to the WT strain (Figure 5). As observed above (Figure 2), the pigment extraction data showing the Co-induced (i) clear decrease in chlorophyll and (ii) (low) decrease in carotenoids and (iii) small increase in phycocyanin, fit better with the whole-cell absorption spectra normalized at 569 nm (a valley of absorbance) than at 800 nm (Figure 5). The Co-elicited strong decline of chlorophyll observed with pigment extraction and whole-cell absorption spectra normalized at 569 nm was confirmed by the Western blot evaluation of the content of the chlorophyll-binding photosystem I reaction center subunit PsaA, which harbors the [4Fe-4S] iron–sulfur cluster F_X_ [56]. The abundance of PsaA was more reduced by Co in the Δ*slr0605*::Sm^R^/Sp^R^ mutant than in the WT strain (Figure 5). In contrast, the content of the PSII reaction center PsbA of both the Δ*slr0605*::Sm^R^/Sp^R^ mutant and the WT strain was not affected by Co (Figure 5).

### 3.5. Cobalt-Elicited Decline of Catalase Activity and Accumulation of ROS Are Exacerbated in the Δslr0605 Mutant as Compared to WT Strain

To study the influence of Slr0605 on the protection against Co, which was shown to trigger oxidative stress in *E. coli* [30], we measured the level of reactive oxygen species (ROS) in WT and Δslr0605 cells, before and after challenging them with Co. The data showed that Co increases the level of ROS more strongly in the Δslr0605 mutant than in the WT strain (Figure 6). Consistently, the activity of the antioxidant enzyme catalase was decreased by Co more significantly in the Δslr0605 mutant than in the WT strain. In contrast, the superoxide dismutase (SOD) activity of the Δslr0605 and WT cells were not affected by Co. Collectively, these results showed that the Co-sensitive Δslr0605 mutant exposed to Co undergoes a decrease in catalase activity and an increase in ROS (Figure 6).

### 3.6. Slr0605 Contributes to Iron Mitigation of Cobalt Toxicity

In WT cells, the Co-induced decline of cell growth and chlorophyll a content was shown above to be attenuated by the presence of extra Fe (Figure 1 and Figure 2). To test whether Slr0605 was involved in this process, we compared the influence of Co and/or Fe supplementation on cell growth and pigment content in the WT strain and the Δslr0605 mutant. The data showed that Fe supplementation did not alleviate the Co-elicited decrease in cell growth and chlorophyll content in the Δslr0605 mutant, unlike what occurred in WT cells (Figure 7).

### 3.7. The Restoration of Cobalt Tolerance in the *Δ*slr0605 Mutant by Expressing the slr0605 Gene, or Its Strep-Tagged Derivative, from a Replicative Plasmid

To prove that the Co sensitivity of Δ*slr0605* cells is due solely to the absence of *slr0605,* this gene was reintroduced in the Δ*slr0605* mutant. For this purpose, *slr0605* was cloned (Section 2) in the pCK replicative plasmid, a Km^R^ derivative of the pFC1 vector, for efficient gene expression from the strong *lambda*-phage *pR*-promoter [34]. Similarly, we also cloned, in pCK, a *slr0605* gene, translationally fused at its 3′OH side to the Strep-tagII, for the facile purification of the Slr0605 protein and to test its activity. The resulting pCKslr0605 and pCKslr0605strep plasmids and the pCK control plasmid were introduced by conjugation [34] in Δ*slr0605*::Sm^R^/Sp^R^ cells, selecting for resistance to Km, Sm and Sp. In all three cases, two Km^R^,Sm^R^/Sp^R^ clones were analyzed by PCR to show that both the pCKslr0605 and pCKslr0605strep plasmids replicate well in the Δ*slr0605* mutant, as does the control pCK vector. The resulting strains, and the WT and Δ*slr0605* control strains, were plated on a solid MM medium with or without 5 µM Co to test their Co tolerance (Figure 8). As expected, the pCKslr0605 and pCKslr0605strep plasmids increased the Co resistance of the Δ*slr0605* mutant up to the WT level. These data show that the reintroduction of the *slr0605* gene, or its Strep-tagged derivative, in the Δ*slr0605* mutant restores its tolerance to cobalt. They also indicate that the strep tag has no negative influence on Slr0605 activity.

### 3.8. Slr0605 Is a Genuine Glutathionyl–Hydroquinone Reductase

In accordance with the KEGG database, where Slr0605 is labeled as a glutathionyl–hydroquinone reductase protein [EC:1.8.5.7], sequence comparison analysis using BLAST [57] indicated that Slr0605 shares homology with Xi-class GSTs (Appendix A). These enzymes catalyze the glutathione (GSH)-dependent reduction in glutathionyl–hydroquinones (GS–hydroquinones) to hydroquinones to avoid the formation of toxic ROS [12,13,14]. Glutathionyl–hydroquinone reductases (GHRs) are widely distributed in bacteria, fungi and plants (not in animals), but have not yet been studied in cyanobacteria. To test the possible GS-HQRs activity of Slr0605, the Strep-tagged Slr0605 protein was extracted from cells propagating the pCKslr0605strep plasmid and affinity purified on Strep-Tactin XT Sepharose (Appendix A). As anticipated, Slr0605 appeared to be a dimeric protein, with a predicted structure similar to GHRs (Figure 9). Consistently, Slr0605 was found to have genuine GHR activity and to be required for tolerance to benzoquinone (Figure 9).

## 4. Conclusions

Glutathione transferases (GST) are widespread detoxication enzymes [1] of great importance for human health [3] and agriculture [5]. However, the analysis of the selectivity/redundancy of GSTs is complex in higher eukaryotes because they possess multiple GSTs as well as various cell types and tissues. In contrast, basic organisms, such as cyanobacteria, are attractive organisms for studying the selectivity/redundancy of GSTs. Indeed, cyanobacteria have fewer GSTs and a simple morphology, especially the unicellular strain *Synechocystis* PCC 6803 [1], which was presently used as a model. Cyanobacteria are also interesting because they are regarded as having evolved GSTs (and other glutathione-using enzymes) for their protection against stresses [1,16].

Among the superfamily of GSTs, the Xi-class enzymes found in prokaryotes, fungi and plants have been studied in a few articles, which reported that they have glutathionyl–hydroquinone reductase activity [12,13,14]. However, the physiological role of Xi GSTs had not yet been well studied, especially in cyanobacteria.

In this study, we report the first analysis of a putative cyanobacterial Xi GST, designated as Slr0605, in the model host *Synechocystis* PCC 6803. We show that Slr0605 is not crucial for standard photoautotrophic growth. However, Slr0605 is required for protection against excess cobalt, a metal crucial in trace amounts for the synthesis of vitamin B12 and other cobalamins [28,29,30], which can become toxic at high doses [31] and are released by industrial pollution [58]. We also show that Slr0605 has a protective effect against the Co-elicited disturbance of photosynthetic pigments, iron (Fe) homeostasis and the biogenesis/repair of enzymatic iron–sulfur [Fe-S] clusters, which trigger oxidative stress. These findings are welcome, since the protective role of GST on cobalt toxicity had not yet been firmly established through the construction and phenotypic analysis of GST-null mutants. We also report that Slr0605 has both the homodimeric form and the glutathionyl–hydroquinone reductase activity of genuine Xi GSTs. In agreement with these findings, Slr0605 appeared to be required for protection against benzoquinone.

## Figures and Tables

**Figure 1 antioxidants-13-01577-f001:**
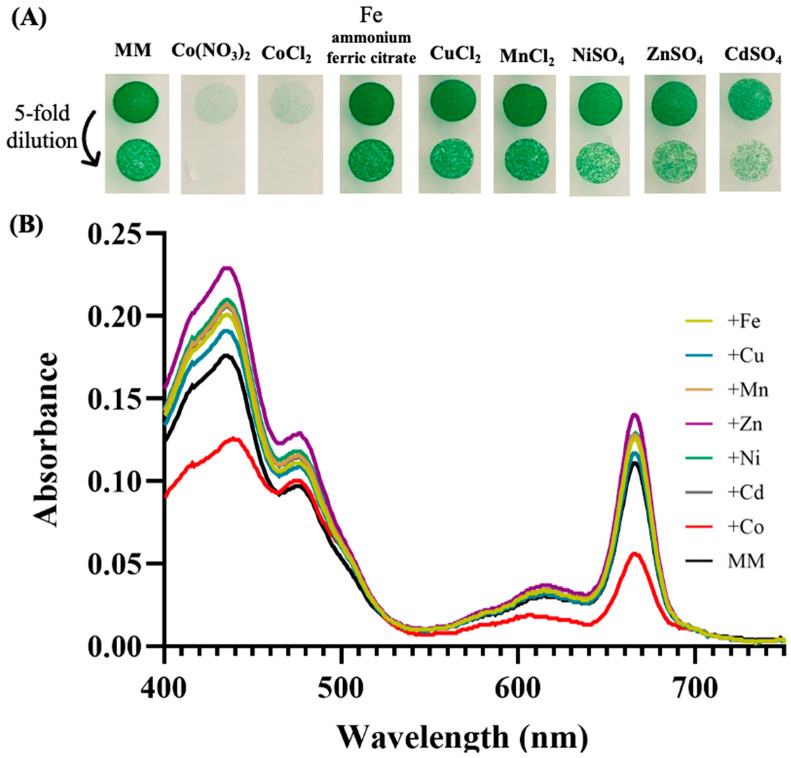
Influence of metals on cell growth and photosynthetic pigments of *Synechocystis*. (**A**) Cells in mid-log phase cultures (OD_750nm_ = 0.5) were spotted as 10 µL dots onto solid mineral medium (MM) with or without 5 µM of the indicated metals. Plates were incubated for 7 days at 30° under standard light C, prior to photography. (**B**) Typical absorption spectra (normalized to light scattering at 800 nm) of the photosynthetic pigments extracted from similar number (1 × 10^7^ cells) of these cells. All experiments were performed at least three times.

**Figure 2 antioxidants-13-01577-f002:**
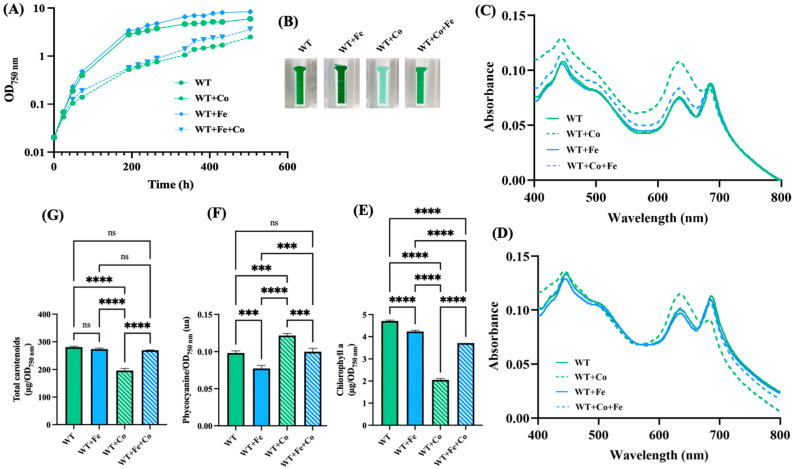
Influence of iron supplementation on cobalt resistance in *Synechocystis*. (**A**) Typical growth curves of cells incubated for the indicated durations under standard light at 30 °C on solid MM with or without either or both CoCl_2_ (5 µM) and FeCl_3_ (153 µM), prior to resuspension in liquid MM and measurements of absorption (OD_750nm_). (**B**) Photographs of spectrophotometric cuvettes containing cells that were incubated for 13 days (312 h) on plates with or without CoCl_2_ (5 µM) and/or FeCl_3_ (153 µM). (**C**,**D**) Typical absorption spectra normalized to light scattering at 800 nm or 569 nm of cells shown in (**B**). (**E**–**G**) Column diagram representation of the quantities of chlorophyll, phycocyanin and carotenoids, respectively, extracted from cells shown in (**B**). All experiments were performed at least three times. Hooks ⊏ indicate significant difference between the two compared experiments (one-way ANOVA test, *p* < 0.0005 or 0.00005, symbolized by *** or ****), while “ns” stands for no statistical difference.

**Figure 3 antioxidants-13-01577-f003:**
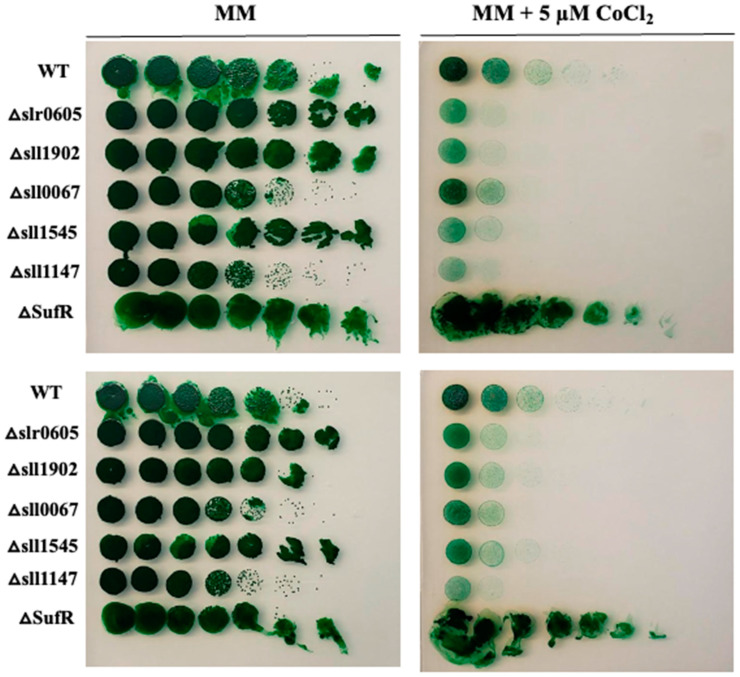
Influence of the deletion of *sufR* on cobalt resistance in *Synechocystis*. Serial five-fold dilution of cultures (OD_750nm_ = 1.0) of the WT strain, the Δ*sufR* mutant and various GST-less mutants (Δ*slr0605*, Δ*sll1545*, Δ*sll1147*, Δ*sll0067* and Δ*sll1902*) were spotted as 10 µL dots onto solid MM medium without or with 5 µM CoCl_2_ and incubated for 13 days at 30 °C under standard light prior to photography. These experiments were repeated at least three times.

**Figure 4 antioxidants-13-01577-f004:**
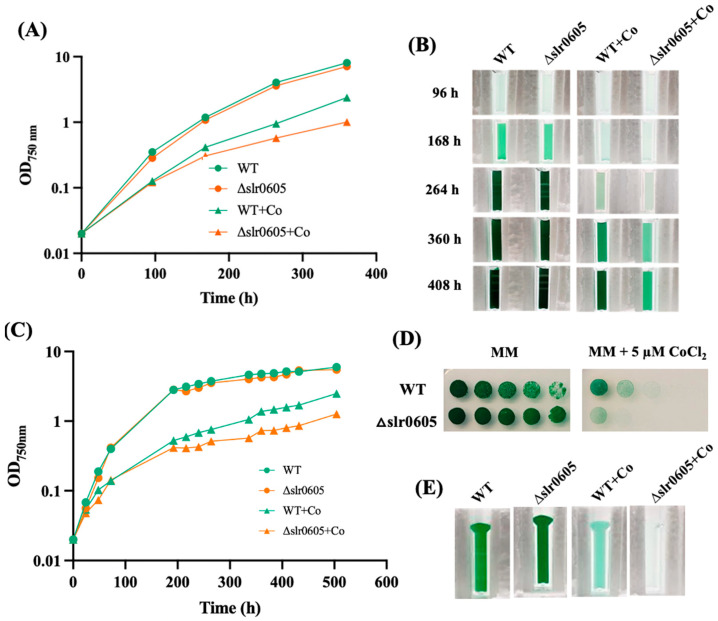
The Δ*slr0605* mutant is hypersensitive to cobalt. (**A**) The typical growth of the WT strain and the Δ*slr0605*::Sm^R^/Sp^R^ deletion mutant in liquid MM with or without 5 µM CoCl_2_. (**B**) At the indicated times, cells were transferred to a spectrophotometric cuvette and photographed. (**C**) Typical growth of the WT strain and the Δ*slr0605*::Sm^R^/Sp^R^ deletion mutant in solid MM with or without 5 µM CoCl_2_. At the indicated times, cells were resuspended in liquid MM and transferred to a spectrophotometric cuvette to measure their absorbance at OD_750nm_ and photographed. (**D**) The serial five-fold dilution of mid-log phase cultures (OD_750nm_ = 0.5) of the WT strain and the Δ*slr0605*:Sm^R^/Sp^R^ mutant, which were spotted as 10 µL dots onto solid MM medium with or without 5 µM CoCl_2_ and incubated under otherwise standard conditions for 13 days. (**E**) These cells were resuspended in liquid MM, transferred to cuvettes and photographed. These experiments were performed at least three times.

**Figure 5 antioxidants-13-01577-f005:**
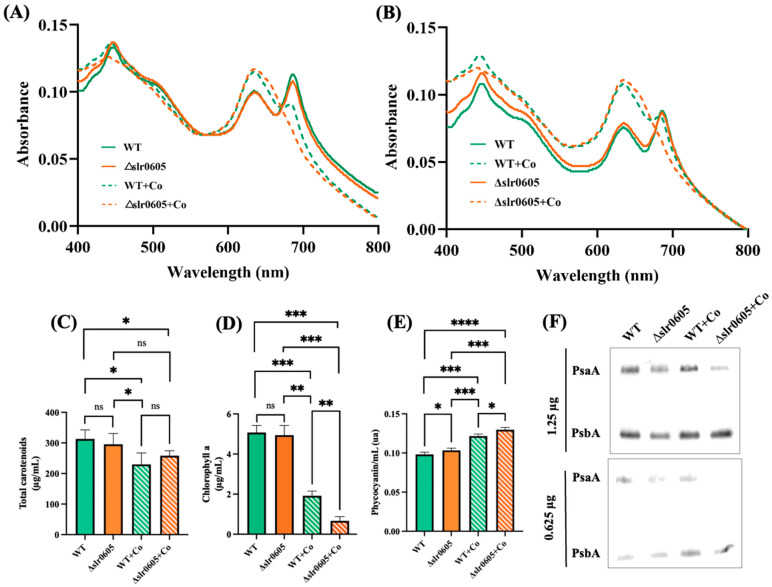
Effect of cobalt on pigment content of wild-type strain and Δ*slr0605*::Sm^R^/Sp^R^ mutant. (**A**,**B**) Whole-cell absorption spectra (normalized to light scattering at 569 nm and 800 nm, respectively) of WT strain and Δ*slr0605*::Sm^R^/Sp^R^ cells grown for 13 days on solid MM with or without 5 µM CoCl_2_. (**C**–**E**) Column diagram representation of quantity of carotenoids, chlorophyll a and phycocyanin of these cells, respectively. (**F**) Western blot analysis of abundance of photosystem I protein PsaA and photosystem II protein PsbA in WT and Δ*slr0605* mutants cultivated in absence or presence of excess Co. Similar quantities of total proteins (either 0.625 or 1.25 μg) were analyzed by Western blots, using antibodies directed against either protein PsaA or PsbA. These experiments were performed three times. Hooks ⊏ indicate significant difference between the two compared experiments (ANOVA test, *p* < 0.05, *p* < 0.005, *p* < 0.0005 or *p* < 0.00005 symbolized by *, **, ***, or ****, respectively); “ns” stands for no statistical difference.

**Figure 6 antioxidants-13-01577-f006:**
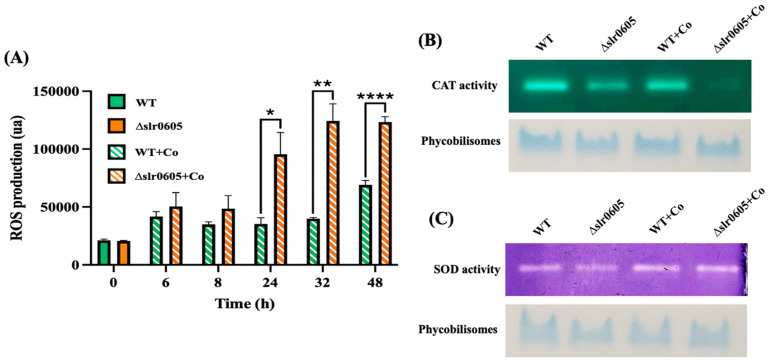
Effect of cobalt on level of Reactive Oxygen Species (ROS) and both catalase (CAT) and superoxide dismutase (SOD) activities in WT and Δslr0605 cells. (**A**) Column diagram representation of ROS levels in WT and Δslr0605 cells before or after incubation with 25 µM Co for indicated durations (ua means arbitrary units). Hooks ⊏ indicate significant difference between the two compared experiments (Welch test, *p* < 0.005, *p* < 0.005 or *p* < 0.00005, are symbolized by *, ** or ****, respectively). (**B**) Catalase activity tested by native PAGE gel-based assay (upper panel) of 10 µg proteins extracted from WT or Δslr0605 cells incubated with or without 5 µM Co for 13 days. As loading control (lower panel), same samples were resolved on 12% polyacrylamide gel showing band corresponding to phycobilisomes, which are naturally colored in blue. (**C**) SOD activity tested by native PAGE gel assay (upper panel) of 10 µg proteins extracted from WT or Δslr0605 cells incubated with or without 5 µM Co for 13 days. As a loading control (lower panel), same samples were resolved on 12% polyacrylamide gel showing blue band corresponding to phycobilisomes. These experiments were performed three times.

**Figure 7 antioxidants-13-01577-f007:**
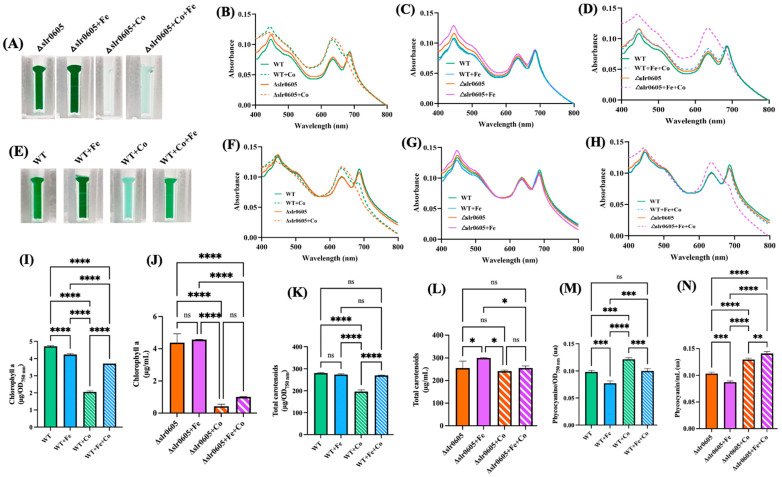
Fe supplementation does not attenuate Co-elicited decline of cell growth and chlorophyll content in Δslr0605 mutant. (**A**,**E**) Appearance of cells grown for 13 days (312 h) ± Co, which were transferred into spectrophotometric cuvettes prior to photography. Typical whole-cell absorption spectra, normalized to light scattering at 800 nm (**B**–**D**) or 569 nm (**F**–**H**) of WT and Δslr0605 cells incubated on solid media with or without extra Co and/or Fe. Column diagram representation of the quantity of the carotenoids (**I**,**J**), chlorophyll a (**K**,**L**) and phycocyanin pigments (**M**,**N**) extracted from these cells. All experiments were performed at least three times. Hooks ⊏ indicate significant difference between the two compared experiments (ANOVA test, *p* < 0.05, *p* < 0.005, *p* < 0.0005 or *p* < 0.00005, symbolized by *, **, *** or ****); ns stands for no statistical difference.

**Figure 8 antioxidants-13-01577-f008:**
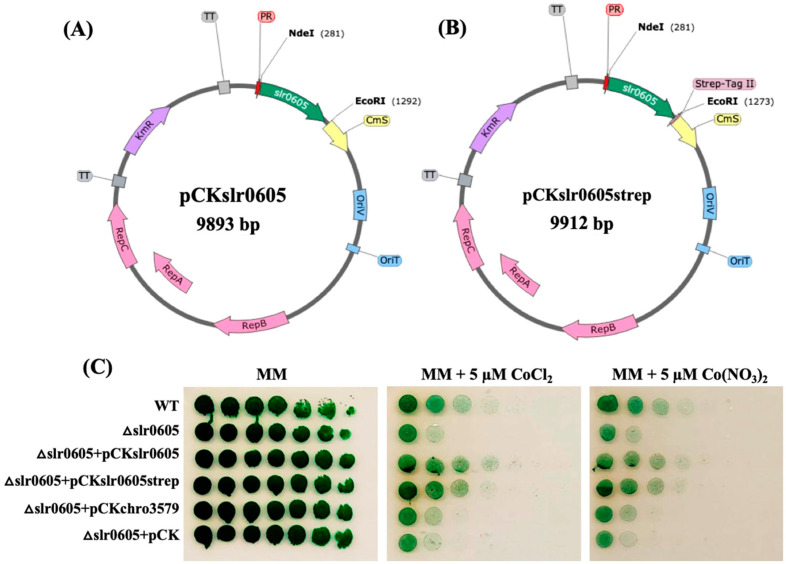
A schematic representation of the pCKslr0605 and pCKslr0605strep plasmids producing Slr0605 with or without a strep tag, and analysis of their influence on Co tolerance of the Δslr0605 mutant. (**A**,**B**) Schematic representation of the pCKslr0605 plasmid and its derivative pCKslr0605-strep for the facile purification of the Slr0605-strep protein. Slr0605 is shown in green, while the antibiotic resistance genes are shown in purple (Km^R^), yellow (Cm^R^, Cm^S^) or dark purple (Amp^R^). The pTwist replicon is shown in grey, while the RSF1010 genes for conjugative transfer and autonomous replication of the plasmids in both *E. coli* and *Synechocystis* are represented in blue rectangles and pink arrows, respectively. The lambda-phage pR-promoter (PR) is shown as the small red triangle. (**C**) The influence of pCKslr0605 and pCKslr0605strep on the Co tolerance of the Δslr0605 mutant. The serial five-fold dilution of cultures (OD_750nm_ = 1.0) of the WT strain or the Δslr0605 mutant and its derivatives, producing Slr0605 with or without a strep tag. All cultures were spotted as 10 µL dots onto a solid MM medium without or with 5 µM Co and incubated for 13 days at 30 °C under standard light, prior to photography. These experiments were repeated at least three times.

**Figure 9 antioxidants-13-01577-f009:**
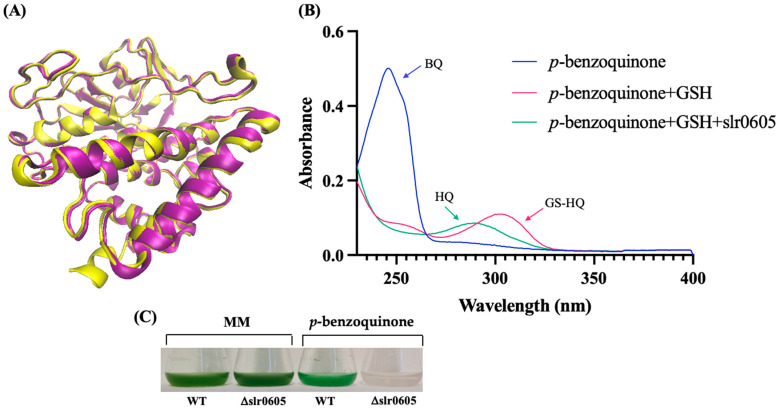
A representation of the structure of Slr0605 and verification of its glutathionyl–hydroquinone reductase activity. (**A**) A superposition of the ribbon diagrams representing the crystal structures of the EcYqjG glutathionyl–hydroquinone reductase (yellow, PDB: 4G0I) and Slr0605 (purple, AlphaFold). The root mean square deviation (RMSD) value pf 2.9 Å indicates that these two proteins share a certain degree of similarity. (**B**) A superposition of the absorption spectra of p-benzoquinone (BQ, blue), S-Glutathionyl–p-hydrobenzoquinone (GS-HQ, red) and HQ (p-benzohydroquinone, green). The addition of glutathione (GSH) to BQ produces GS-HQ, a stable compound that is converted into HQ by Slr0605 within a few minutes. (**C**) Typical growth of the WT strain and the Δslr0605::Sm^R^/Sp^R^ deletion mutant in liquid MM with or without 14 µM p-benzoquinone.

## Data Availability

Data are contained within the article and Appendix A.

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
