# Peer review of "First Characterization of a Cyanobacterial Xi-Class Glutathione S-Transferase in Synechocystis PCC 6803"

_antioxidants, 2024, doi:10.3390/antiox13121577_

Round 1
Reviewer 1 Report
The authors of the manuscript focused on the characterization of an Xi GST in cyanobacteria.
The authors already published some data regarding this type of enzymes in cyanobacteria. The topic is of interest and here a new candidate was described in detail. The experiments were carried out in detail and the results give inside into the role of this enzyme in protection against Co.
However, some minor changes are needed and some rewriting is necessary.
One larger issue is the writing part of the results, a lot of the material method section appears here (line 267-269; sec 3.4 first paragraph; sec 3.7 first paragraph) and also some explanation, which should be in the discussion was written in this part (examples line 215-218: line 229-233). Here some restructuring is needed.
1. The authors should reduce the amount of self-citation, since the are over 26% (allowed are 15%) of all citations are from this group
2. The authors should check the style of the citations (in the reference list and in the text (mix of italics, bold and normal))
3. Line 252: this is not a histogram but a column diagram
4. Line 287: to be published elsewhere is not very common, change to unknown
5. Figure 5: one experiment should be enough.
6. Figure 4: it is very striking that growth on solid and liquid media is very similar. Is this correct?
7. Figure 5: western blot: is the same amount of protein or the same number of cells is loaded to the gel: this should be also mentioned in the M&M
8. Figure 8: the description of the growth experiment is missing. And the construction part can be moved into the M&M or SI material.
Reviewer 2 Report
The paper describes the characterisation of a Glutathione-S-transferase (GST) of the cyanobacterium Synechocystis PCC 6803, encoded by the Slr0605 gene, through the analysis of deletion/complementation mutants in response to stress conditions induced by metal (cobalt) excess. The effect of iron supplementation and of the constitutive activation of the Suf machinery, responsible for the biogenesis of Fe-S clusters, is also presented and a role for Slr0605 in protecting Synechocystis cells from cobalt toxicity is suggested. In addition, the enzymatic activity of the isolated Slr0605 protein is shown to be consistent with its belonging to the xi-class of GSTs.
The study is interesting as it contributes to elucidating some aspects related to metal stress and resistance in cyanobacteria involving GSTs.
There are however a few issues that should be addressed before publication in order to improve the comprehension of the results and the readability of the manuscript, as explained below.
Materials & Methods:
- Line 74: two different light intensities are listed but only standard light was used for cyanobacteria growth, please check.
Results:
- Figure 1B: the spectra show a significant decrease in absorbance when cells are grown in the presence of cobalt. It is however not clear if this is simply due to a growth impairment (i.e. a lower number of cells on the plate and therefore less pigments to be extracted) or to a reduced synthesis/enhanced degradation of pigments in a single cell or both. If possible, pigment content should be expressed on a cell-number basis and absorption spectra normalised (in the red and/or in the blue region of the spectrum) to better highlight any differences in pigment relative amount/composition.
- L223/224: “Typical absorption spectra (corrected for the absorbance at 800 nm) of the …”
- L235-241: The observed discrepancy is most probably the result of light scattering from the sample. If not acquired using compensation methods (opal glass diffusers/integrating sphere), as in this case, whole cells absorption spectra are inevitably affected by the light scattering of the cell suspension, the contribution of which is not uniform across the spectrum and higher at shorter wavelengths. This, together with the low absorbance values of the samples, make spectra normalisation/analysis cumbersome, as some differences are due to scatter-induced distortions and not to intrinsic absorption. It is therefore suggested to limit the analysis and the discussion to the methanol extracts, which are in this sense more reliable. The same holds true for the whole cells absorption spectra of figure 2C/D (as well as those of Figs 5A/B and 7B-D 7F-H).
- Sections 3.3 and 3.4: The detailed description of plasmids construction should be moved to the “M&M” section, and Figure 8A/B should be enlarged or moved to the Supplementary Materials. Please also check table S2 as there are some inaccuracies/missing information.
- Section 3.4 and Figure S2: Validation of the mutant lacking sufR. Control PCR analysis of the wild type is missing. PCR1 fragment is 520 or 580 bp (in the main text is written 520 bp)?
- It is suggested to rename section 3 as “Results and Discussion” and section 4 as “Conclusions”
L487: Figure S6 is Figure 9 main text.
Round 2
Reviewer 1 Report
can be published
can be published